# Calculated hydration free energies become less accurate with increases in molecular weight

Stefan M. Ivanov [ID] *

Faculty of Pharmacy, Medical University of Sofia, Sofia, Bulgaria

* sivanov@ddg-pharmfac.net

## Abstract

In order for computer-aided drug design to fulfil its long held promise of delivering new medicines faster and cheaper, extensive development and validation work must be done first. This pertains particularly to molecular dynamics force fields where one important aspect–the hydration free energy (HFE) of small molecules–is often insufficiently analyzed. While most benchmarking studies report excellent accuracies of calculated hydration free energies–usually within 2 kcal/mol of experimental values–we find that deeper analysis reveals significant shortcomings. Herein, we report a dependence of HFE prediction errors on ligand molecular weight–the higher the weight, the bigger the prediction error *and* the higher the probability the calculated result is erroneous by a large amount. We show that in the drug-like molecular weight region, HFE predictions can easily be off by 5 kcal/mol or more. This is likely to be highly problematic in a drug discovery and development setting. We make our HFE results and molecular descriptors freely and fully available in order to encourage deeper analysis of future molecular dynamics results and facilitate development of the next generation of force fields.

**Data Availability Statement:** All relevant data are within the manuscript and attached figures and S1 File.

**Funding:** This study is financed by the European Union-NextGenerationEU through the National Recovery and Resilience Plan of the Republic of

## Introduction

For decades, computer-aided drug design (CADD) has held the promise of greatly accelerating drug discovery and bringing much needed pharmacotherapeutic solutions to patients and medical professionals. CADD practitioners like to point to success stories such as the development of raltegravir–an HIV integrase inhibitor [1]–and doramapimod–a highly potent kinase inhibitor–where modeling and simulation have been instrumental in bringing about new therapeutic agents [2]. In the case of raltegravir, all-atom molecular dynamics (MD) simulations [3] revealed a cryptic trench in HIV integrase and led to the discovery of a new class of antiretrovirals–the integrase inhibitors [2]. Similarly, explicit solvent MD simulations revealed a 10 Å shift of the F169 side chain of p38 MAP kinase, exposing a cryptic pocket in the presence of BIRB 796 –a novel ligand, later named doramapimod—which was evaluated by Boehringer Ingelheim in clinical trials for the treatment of inflammatory diseases [4]. As it samples and explores different conformational states, molecular dynamics is particularly well suited to

Bulgaria, project BG-RRP-2.004-0004-C01. The in silico calculations were performed at the Centre of Excellence for Informatics and ICT, supported by the Science and Education for Smart Growth Operational Program and co-financed by the European Union through the European Structural and Investment Funds (Grant No. BG05M2OP001-1.001-0003). The funders played no role in the study design, data collection and analysis, decision to publish, or preparation of the manuscript.

**Competing interests:** The author has declared that no competing interests exist.

identifying cryptic pockets which remain occluded in many crystal structures [5,6]. While raltegravir and doramapimod are excellent examples of molecular modeling and dynamics delivering new and actionable knowledge that leads to new drugs or drug-like molecules, there are hundreds of types of cancer [7] accounting for millions of deaths yearly [8]. Clearly, much remains to be desired. Indeed, while the rapid turnaround *in silico* hit discovery [9] provides and the unraveling of biomolecular association [10–16] mechanisms are certainly beneficial to drug design and discovery, there is no shortage of diseases and malignancies in need of new vaccines and treatments. The lack of vaccines for malaria [17] and hepatitis C [18], despite the decades of research, are two prominent examples. Moreover, even in certain cancers where there are therapeutic options available, such as pancreatic cancer, the survival rate is still very low [19], further exacerbating the need for new antineoplastic molecules [20].

One of the enabling–or limiting–factors in the utility of computer modeling and all-atom molecular dynamics simulations [5,21] for the purposes of drug design and discovery is the fidelity and accuracy of the potentials used in these campaigns. On the one end of the computational spectrum, we have molecular docking [22]—a popular and conceptually simple technique used to rapidly screen libraries of ligands against a target of interest. Docking typically employs unsophisticated, easily computable empirical or knowledge-based potentials [23] to evaluate different ligand conformations and rank all the ligands in a library. On the other extreme, we have the theoretically rigorous and computationally demanding absolute binding free energy calculations (ABFEs) [24]. Typically, docking tries to discern ligands that bind the target of interest from ones that do not by rapidly evaluating simple scoring functions while ABFEs try to accurately calculate, at least in principle, the absolute free energy of binding ($\Delta G_{bind}$ or simply $\Delta G$) between the target and every ligand being screened by covering the relevant conformational space and evaluating $\Delta G_{bind}$ using potential energy functions referred to as force fields. In principle, $\Delta G_{bind}$ is the difference between the free energy (G) of the bound state and the free energy of the unbound protein and unbound ligand in solution. In practice, ABFEs do not calculate absolute free energies (Gs) but only free energy changes or differences ($\Delta G$s) by constructing a thermodynamic cycle and decoupling the ligand from the bound state and from solvent and taking the difference of these two terms [25]. The free energy change upon decoupling the ligand from solvent, i.e. desolvating the ligand, is the desolvation free energy or the solvation free energy multiplied by -1. In the case where the solvent is water, what is being calculated is the (de)hydration free energy (HFE). In contrast to ABFEs, where the ligand is decoupled from the system, in relative binding free energy (RBFE) calculations, the ligand is transformed into a different ligand, typically a close analog, in order to calculate the relative change in free energy ($\Delta\Delta G$) between the two [26–28].

For a chemical compound to be a viable drug in clinical practice, its hydration free energy must be within a suitable range. If a molecule that is too hydrophilic is administered orally, e.g. as a tablet, it will remain mostly in the fluid inside the gastrointestinal tract and fail to distribute throughout the body [29]. Conversely, if a molecule that is too hydrophobic is administered orally, it will also fail to reach its intended target, as it fails to pass into the intestinal fluid [30]. The hydration free energy is a determinant of a drug molecule's liberation from its pharmaceutical formulation and its absorption and distribution in the body, which are the subject of an entire scientific discipline–pharmacokinetics [31]. Moreover, highly lipophilic molecules are much more likely to be promiscuous binders and tend to behave as pan-assay interference compounds (PAINS) [32]. Further still, highly hydrophilic or hydrophobic compounds are often difficult to work with in terms of chromatography analysis, purification [33], and formulation [34]. Given the importance of HFEs in the development of pharmaceuticals, much effort has been devoted to developing methods for their rapid and accurate prediction [35,36], as evidenced in the SAMPL challenges [37].

($\Delta$)$\Delta G$ can be computed numerically via thermodynamic integration (TI [38]). In TI, the free energy difference between the two states of interest (also referred to as end states) is evaluated by connecting them through some functional form f($\lambda$) of a nonphysical coordinate $\lambda$, calculating the derivative of the potential with respect to $\lambda$ ($\partial U/\partial \lambda$), and estimating the area under the $\partial U/\partial \lambda$ versus $\lambda$ curve [39–41]. TI makes use of the malleability of the potential energy function which can be calculated for the nonphysical, intermediate, mixed states, which is why it belongs to a class of free energy methods referred to as alchemical transformations. In this way, one ligand can be transmuted into another and the free energy difference between the two can be calculated. The transformation can be done by modifying van der Waals and electrostatic interactions simultaneously (the so-called one-step approach) or separately (the two-step approach) [42]. In the simplest case, f($\lambda$) = $\lambda$, which is referred to as linear scaling [43]. To avoid singularities in the potential energy function when particles appear or disappear, i.e. are being inserted into or decoupled from the system, a modified or softcore potential is used [44].

Clearly, the quality of the force fields is one of the major factors in determining the outcomes in such campaigns. The calculated hydration free energy of a ligand is fully determined by two force fields: the water model and the small molecule force field, and their interplay. Over the past several decades, literally hundreds of water models have been proposed and developed, yet in biomolecular modeling, the rigid, fixed-charge, 3-point TIP3P [45] model remains among the most widely used, if not the most widely used [46], despite its many known shortcomings [47]. Similarly, the general Amber force field (GAFF [48]) and its subsequent refinements hold this position in simulating drug and drug-like molecules. Other notable families of force fields are CHARMM [49], the Open Force Field (OpenFF) [50], and OPLS [51]. More recently, a new 4-point rigid water model named OPC [52] was developed by optimizing the point charge distribution in water that reproduces its bulk properties far more accurately than TIP3P; a 3-point version named OPC3 [53] was developed as a compromise between speed and accuracy. It was hoped that an improvement in bulk property reproduction would also translate into an improvement in hydration free energy predictions. For small sets of organic molecules, the new water models did indeed outperform TIP3P [52,54]. However, those studies use only a small number of ligands to carry out the validation and do not report how they select the molecules they use out of the 642 available for benchmarking from the FreeSolv data set [55]–the largest and most comprehensive curated set of experimentally measured hydration free energies.

Here, we report a thorough analysis of the performance of hydration free energy calculations on the entire FreeSolv set for a combination of the GAFF2.11 force field with the TIP3P, OPC, and OPC3 water models, and compare and contrast all-atom molecular dynamics hydration free energy calculations to the more affordable two-dimensional quantitative structure–activity relationship (2D QSAR [56]) approach based on low-level molecular descriptors [57]. We discuss their merits and shortcomings from the standpoint of computer-aided drug discovery and development. Moreover, we also compare the performance of GAFF2.11 with the previous version of GAFF– 1.81 –demonstrating a robust means of validating edits to force fields that is far more relevant to drug discovery, design, and development than what physics focused research laboratories typically employ. We demonstrate that calculated HFEs become increasingly inaccurate with increases in ligand molecular weight and the number of rotatable bonds. Finally, we show that overall statistics like the coefficient of determination ($R^2$) and root mean square error (RMSE) in benchmarking studies can often be misleading and overshadow significant shortcomings. We demonstrate that deeper analysis is needed to reveal and interpret these shortcomings and showcase such an analysis for HFE calculations on the FreeSolv set.

## Methods

### System setup for thermodynamic integration simulations

To generate starting structures for our simulations, we used the sdf-format ligand structures as provided by FreeSolv. All 642 neutral FreeSolv ligands were processed with *antechamber* from Amber18 to generate Amber-compatible mol2 files with the appropriate atom types and AM1_BCC partial charges [58]. All ligands were parameterized with GAFF versions 2.11 and 1.81. GAFF2.11-parameterized ligands were solvated with *tleap* in cubic boxes of TIP3P, OPC, and OPC3 water with a wall distance of 24 Å; GAFF 1.81-parameterized ligands were solvated with TIP3P water only.

### Simulation protocol for thermodynamic integration simulations

The solvated systems were subjected to 2000 steps of energy minimization. The systems were then heated from 100 to 300 K over a period of 100 ps at constant volume, followed by 100 ps of density equilibration, 100 ps of constant pressure and temperature (NPT) equilibration, and were finally subjected to NPT production runs of 250 ps under 1 bar, 300 K with periodic boundary conditions. Constant temperature and pressure were maintained with the the Langevin thermostat [59] and Berendsen barostat [60], respectively. Collision frequencies for temperature coupling were 2 ps$^{-1}$; the pressure relaxation time was set to 2 ps. A cutoff of 12.0 Å was used for nonbonded interactions. Long-range electrostatics beyond the real space cutoff were computed with the particle-mesh Ewald (PME) scheme [61]. The time step was set to 1 fs to keep the simulations stable during alchemical transformations (decoupling); bonds to hydrogen were not constrained.

### Thermodynamic integration hydration free energy calculations

Simulations were carried out with one-step thermodynamic integration with softcore potentials and linear scaling [42,44] using the *pmemd.cuda* MD engine from Amber18 [62]. Ligands were decoupled from the solvation box in 21 evenly spaced λ-windows ranging from 0.0 to 1.0 in intervals of 0.05. The *scalpha* and *scbeta* parameters, which control the softness of the potential, were set to 0.5 and 12 Å$^2$, respectively, as in previous work [12]. To avoid any Hamiltonian lag [63], energy minimization, heating, density equilibration, NPT preproduction equilibration, and production dynamics were all carried out with potential energy functions, corresponding to the λ-value of every λ-window. Hydration free energies, error estimates, and convergence metrics [64,65] were computed from the decorrelated dV/dl values (the derivative of the potential with respect to λ) from the Amber output files with the *alchemlyb* [66] TI estimator.

### 2D QSAR hydration free energy calculations

Molecular descriptors were calculated with *RDKit* as described previously [12]. Descriptors were also normalized by molecular weight; parameters with zero variance were removed from consideration, leaving around 280 descriptors in total, which were scaled from 0 to 1. We then trained a Gaussian kernel support vector regressor (SVR [67,68]) with scikit-learn [69] on the experimental hydration free energies. The 642 ligands were split into a training and test set in a 70:30 ratio; the scaled descriptors (the independent variables) and hydration free energies (the dependent variables) for the training compounds were used to build an SVR model to predict hydration free energies; the model was then tested against the HFEs from the test set. To choose the optimal model hyperparameters, during training we performed a grid search with 5-fold crossvalidation for the regularization parameter $C$ varying it from $10^{-9}$ to $10^9$ in 10-fold

increments and the kernel coefficient *gamma* using 'scale', 'auto', 0.1, 0.2, 0.3, 0.4, 0.5, 0.6, 0.7, 0.8, 0.9, 1.0, 5, 10, 15, and 25 as possible options. The best performing model was then used to predict the HFEs of the test set ligands and the prediction errors ($\Delta G_{experimental} - \Delta G_{calculated}$, in kcal/mol) and relative errors ($\Delta G_{calculated}*100/\Delta G_{experimental}$, in per cent (%)) were recorded. This procedure was repeated 10,000 times for random train/test splits, all in a 70:30 ratio.

## Results

### Thermodynamic integration hydration free energy calculations

The hydration free energy calculations using the TIP3P water model and GAFF2.11 force field resulted in a pleasing $R^2$ of 0.84 and an RMSE of 1.78 ± 0.05 kcal/mol when compared to experiment (Fig 1).

Somewhat surprisingly, the OPC model produced a lower $R^2$ and a higher RMSE ($R^2$ = 0.67, RMSE = 2.54 ± 0.05 kcal/mol), whereas OPC3 performed similarly to TIP3P ($R^2$ = 0.81, RMSE = 2.18 ± 0.07 kcal/mol). Notably, there is near perfect agreement between the 3-point water models TIP3P and OPC3 ($R^2$ = 0.97, RMSE = 0.97 ± 0.07 kcal/mol, S1 Fig), far more than between OPC and OPC3 ($R^2$ = 0.79, RMSE = 2.02 ± 0.06 kcal/mol). It is also notable that the majority of points lie above the x = y identity line for all three water models. Consequently, the density distribution curves for the prediction errors or residuals ($\Delta G_{experimental} - \Delta G_{calculated}$) are left-shifted from the x = 0 kcal/mol line (Fig 1D). In the case of a near-perfect prediction, the density distribution curve is tall and narrow, tightly centered around x = 0 kcal/mol; the poorer the predictions, the lower the peak and the wider the distribution. Among the three water models, TIP3P has the tallest and narrowest error distribution. Again, the distribution for the 3-point OPC3 is very similar in shape, only slightly shorter and wider, whereas the OPC distribution is appreciably different with larger errors (residuals) being much more prevalent than with TIP3P and OPC3 where the majority of predictions fall within 2 kcal/mol of the experimental values (the region between the dashed black lines in Fig 1D).

Moreover, we analyze prediction accuracy in terms of relative errors given in percentages as $\Delta G_{calculated}*100/\Delta G_{experimental}$. This is important as it allows clearly and easily identifying cases where the workflow mispredicts the sign of the hydration free energy (i.e. HFE has been estimated to be positive when it is on fact negative or vice versa), which is a more substantial misprediction than merely getting the value wrong by a certain amount. Such points lie to the left of the x = 0% axis (identified with a solid red line in Fig 1E). In the case of a perfect predictor, all data points would stack at x = 100% (solid black line in Fig 1E), i.e. the prediction perfectly matches experiment in sign and in magnitude. For x > 0% and x ≠ 100%, the predictor has correctly estimated the sign of the HFE, but not its magnitude, whereas for x < 0%, the HFE sign has been mispredicted. In terms of relative errors, the three water models have much more similar distributions with the three modes lying close to each other. Again, TIP3P has the tallest and narrowest distribution, with the second tallest this time being OPC rather than OPC3, the latter two being very similar in shape and position. The TIP3P curve also has the smallest area lying to the left of x = 0%, i.e. TIP3P has the highest number of correctly predicted HFE signs (574/642 or 89%), followed by OPC (547/642 or 85%) and OPC3 (530/642 or 83%).

Further, we present the error distributions as a function of molecular weight (Fig 2A). We see that the molecular mechanics-based predictions exhibit a rightward fanning out of the error distributions, i.e. errors become larger as molecular weight increases. Moreover, we stress that not only does the distribution become wider to the right, the *proportion* of errors within the -2 to 2 kcal/mol region between the dashed black lines becomes smaller and smaller as one moves to the right of the x axis. Near the beginning of the x axis, the majority of

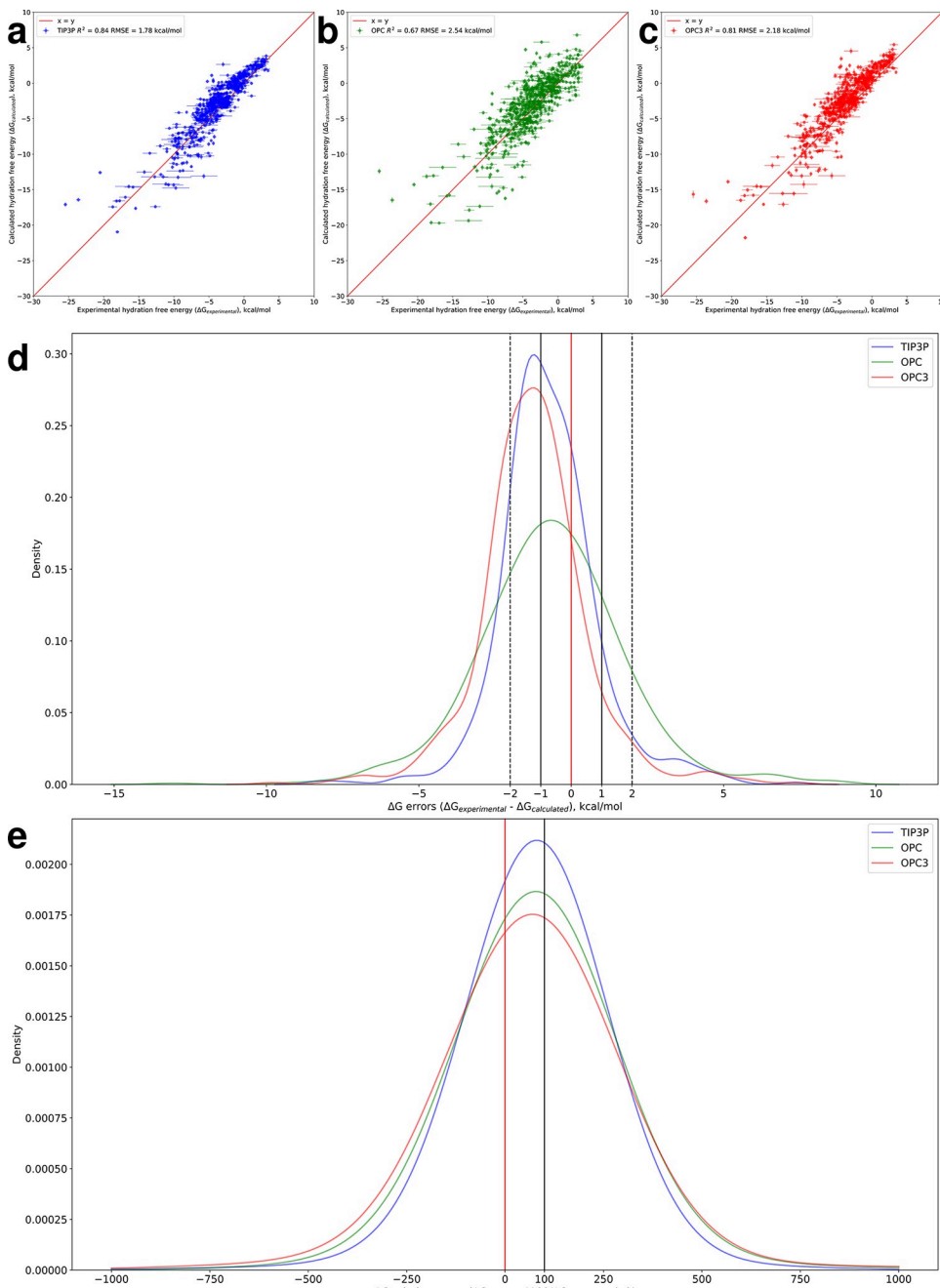

**Fig 1. Calculated versus experimental free energies. a** Scatter plot of calculated versus experimental free energies for GAFF2.11 with TIP3P water. The coefficient of determination ($R^2$) and root mean square error (RMSE) are given in the figure legend. Error estimates for experimental and calculated HFEs are given as horizontal and vertical error bars, respectively (note that most of the calculated error estimates are too small to see at this scale). **b** Scatter plot of calculated versus experimental free energies for GAFF2.11 with OPC water. The $R^2$ and RMSE are given in the figure legend. Error estimates for experimental and calculated HFEs are given as horizontal and vertical error bars, respectively (note that most of the calculated error estimates are too small to see at this scale). **c** Scatter plot of calculated versus experimental free energies for GAFF2.11 with OPC3 water. The $R^2$ and RMSE are given in the figure legend. Error estimates for experimental and calculated HFEs are given as horizontal and vertical error bars, respectively (note that most of the calculated error estimates are too small to see at this scale). **d** Error ($\Delta G_{experimental} - \Delta G_{calculated}$) distributions for the three water models The x = 0 kcal/mol axis indicates the position of perfect predictions; the regions between the vertical solid and dashed lines indicate the location of predictions within 1 and 2 kcal/mol of experimental values, respectively. **e** Relative error ($\Delta G_{calculated}*100/\Delta G_{experimental}$) distributions for the three water models. The x = 100% axis indicates the position of perfect predictions; the x = 0% axis separates ligands

whose calculated HFE is of the correct sign (right of the axis) from ligands whose HFE sign has been mispredicted (left of the axis).

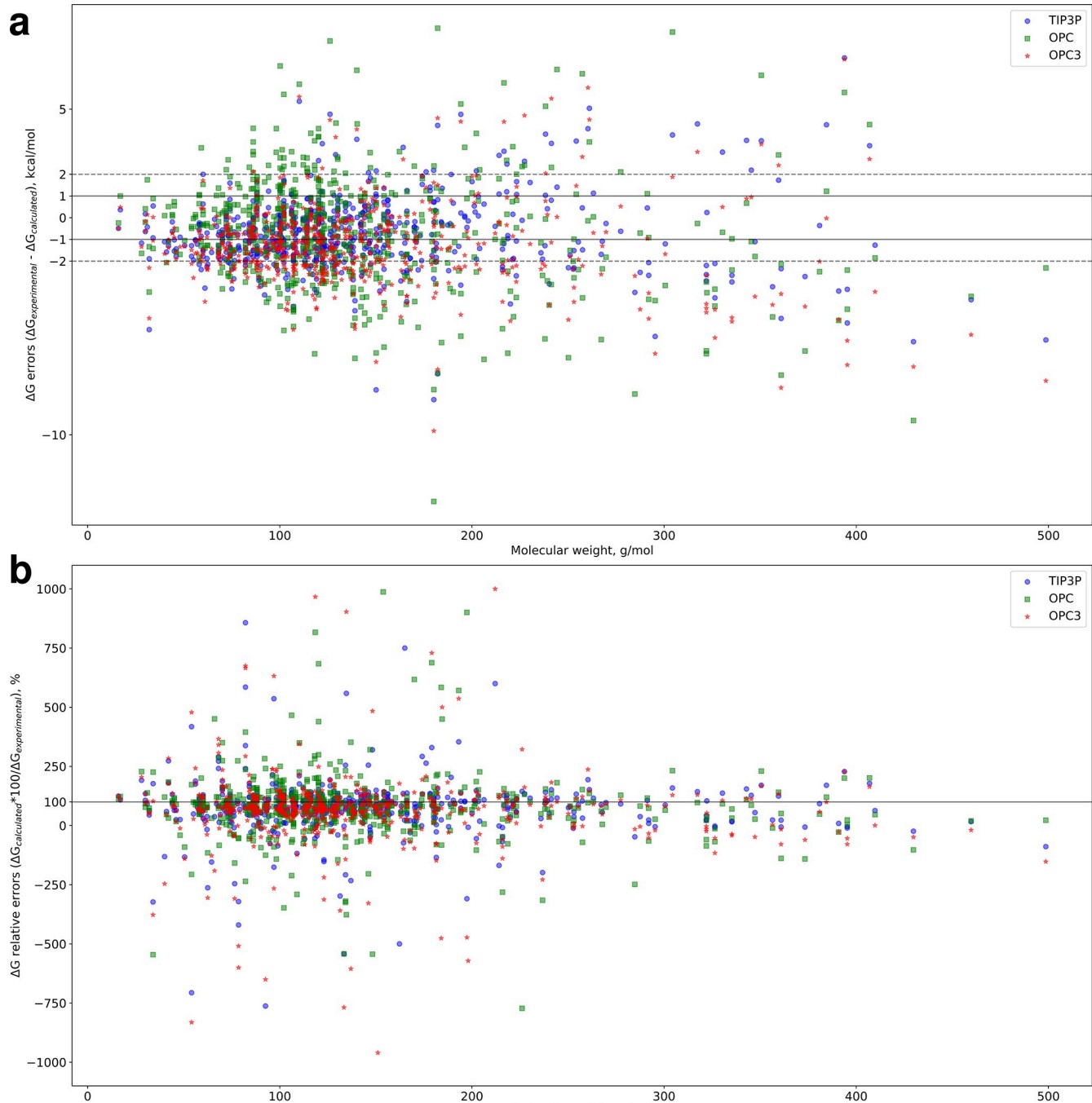

**Fig 2. Hydration free energy prediction errors versus molecular weight. a** GAFF2.11 HFE errors ($\Delta G_{experimental} - \Delta G_{calculated}$) for the three water models plotted against molecular weight. The regions between the horizontal solid and dashed lines indicate the location of predictions within 1 and 2 kcal/mol of experimental values, respectively. **b** GAFF2.11 HFE relative errors ($\Delta G_{calculated} * 100 / \Delta G_{experimental}$) for the three water models plotted against molecular weight. The y = 100% axis indicates the position of perfect predictions.

predictions lie within 2 kcal/mol of experimental values. Moving to the right, such predictions become the minority. This means that not only do errors become larger with increasing molecular weight, large errors become more likely with increasing molecular weight. Interestingly, an opposite trend appears when examining relative errors as a function of molecular weight (Fig 2B)–the distribution is wider on the left of the x axis, up to around 200 g/mol, and narrows after that. However, relative errors behave similarly to absolute errors in that smaller residuals are more prevalent in the low-molecular-weight region of the plot and become less likely in the high-weight region.

For the TIP3P + GAFF2.11 combination, we performed energy minimization, heating, density equilibration, preproduction equilibration, and production dynamics in four independent replicas. Overall results for the hydration free energy estimates, computed from the production stages of the four replicas, are nearly indistinguishable–the 4 replicas agree with each other up to an $R^2$ = 0.99 (S2 Fig). Finally, we compare the latest version of GAFF– 2.11 –to version 1.81 using the TIP3P water model. We find a minor difference in performance–GAFF2.11 produced an $R^2$ of 0.84 and an RMSE of 1.78 ± 0.05 kcal/mol; GAFF1.81 produced and $R^2$ of 0.88 and an RMSE of 1.70 ± 0.05 kcal/mol with a slightly narrower error distribution (S3 Fig).

## 2D QSAR hydration free energy calculations

We now examine the performance of the support vector machine (SVM) machine learning (ML) models on HFE predictions. We generated 10,000 random train/test splits in a 70:30 ratio which means that each split has 193 ligands in the test set and we have 1,930,000 predictions in total. We generated multiple models from different train/test splits to gauge the performance of such models much more reliably than if we were to use a single model. Overlaying the SVM and GAFF2.11 + TIP3P error distributions plotted against molecular weight highlights the difference in performance (Fig 3). The machine learning models have a narrow, symmetric error distribution tightly centered around y = 0 kcal/mol and y = 100% for the absolute and relative errors, respectively. Notably, the error distributions from the ML predictions do not get wider to the right of the molecular weight axis. This contrasts starkly with the TIP3P ΔG error distribution which becomes wider to the right; conversely, TIP3P relative errors tend to be largest around a molecular weight of 100 g/mol and become narrower to the extremes of the weight range. In contrast to the ML ΔG errors which have a narrow, symmetric, elongated distribution centered on the y = 0 kcal/mol (perfect prediction) axis, TIP3P ΔG errors are downshifted and centered around y = -1 kcal/mol. Similarly, the TIP3P relative error distribution is much wider than its ML counterpart, albeit not as downshifted from the y = 100% (perfect prediction) axis as the ΔG error distribution. Crucially, the ML models mispredicted the signs of the HFEs in around 2% of cases compared to around 11% for the TI-based workflow with TIP3P water, and around 15 and 17% with OPC and OPC3 water, respectively.

Notably, some of the descriptors show significant correlation or anticorrelation (given here as the correlation coefficient R) with the error from HFE calculations (all *RDKit* descriptors and HFEs are given in S1 File; descriptor definitions can be found in the *RDKit* documentation and the references therein). When looking at signed errors from the TIP3P calculations, the descriptor with the strongest anticorrelation to error is the VSA_Estate3 molecular surface descriptor [70] normalized by molecular weight (VSA_Estate3/MolWt, R = -0.38). Conversely, the molecular surface descriptors PEOE_VSA14 and SlogP_VSA3 [71] have the strongest positive correlation to HFE error (R = 0.41 and R = 0.46, respectively). The strongest correlations between descriptors and HFE errors, however, appear when taking the absolute values of the errors. In this case, the descriptor with the largest correlation is the number of heteroatoms (R = 0.53) followed by the VSA_Estate3 molecular surface descriptor (R = 0.46).

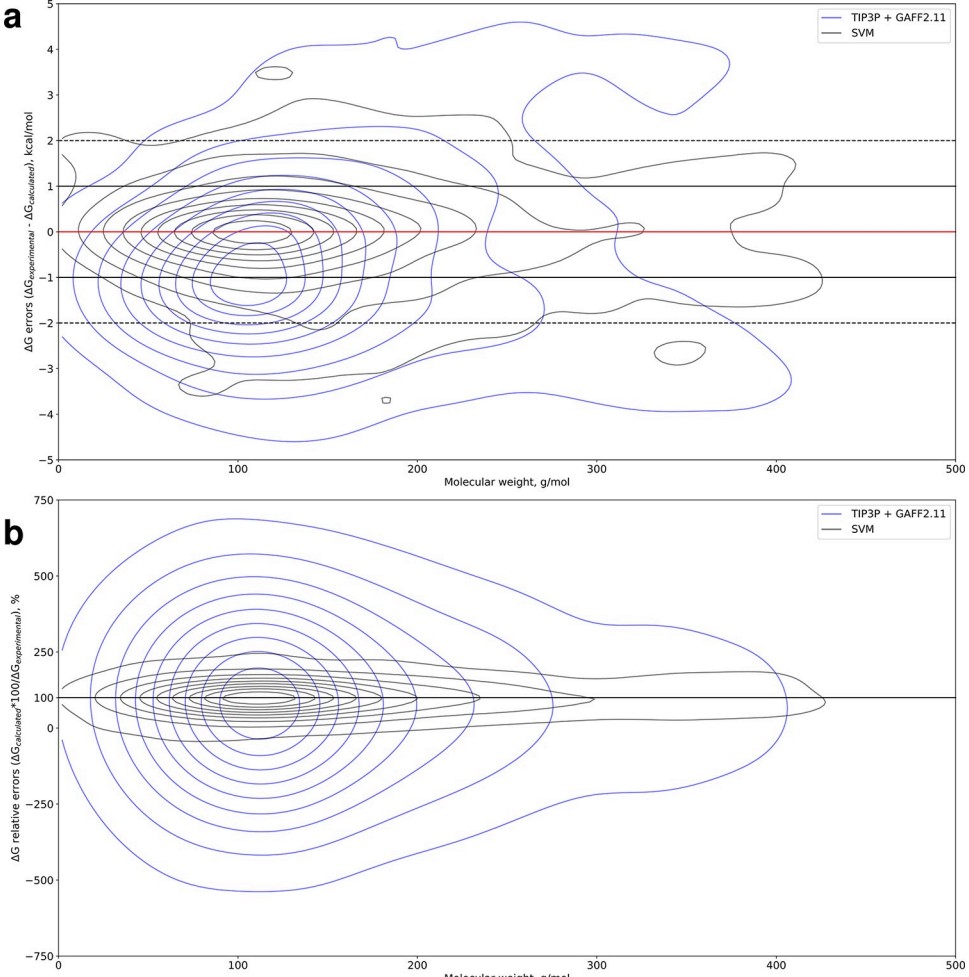

**Fig 3. Hydration free energy prediction errors versus molecular weight. a** Bivariate kernel density estimate plot of HFE errors ($\Delta G_{experimental} - \Delta G_{calculated}$) for GAFF2.11 with the TIP3P water model versus support vector machine (SVM) predictors against molecular weight. The regions between the horizontal solid and dashed lines indicate the location of predictions within 1 and 2 kcal/mol of experimental values, respectively. **b** Bivariate kernel density estimate plot of HFE relative errors ($\Delta G_{calculated}*100/\Delta G_{experimental}$) for GAFF2.11 with the TIP3P water model and SVM predictors against molecular weight. The y = 100% axis indicates the position of perfect predictions.

## Discussion

We report a thorough, rigorous benchmark of the TIP3P, OPC, and OPC3 water models with the popular general Amber force field for small molecules. Due to a limitation on the computational resources available to us, we present results based on fairly limited sampling–for every ligand, the HFE estimate is based on 5.25 ns of production dynamics (21 windows x 0.25 ns of sampling per window). Admittedly, this is not sufficient in duration for our automated pipeline to achieve convergence for every ligand. However, as we compare different water models under identical conditions, we can still make fair and valid comparisons and draw meaningful, useful conclusions. In S4 Fig, we plot the distributions of the A_c convergence metric [64,65] from the simulations using the TIP3P, OPC, and OPC3 water models. An A_c value of 1 indicates perfect convergence across the 21 λ windows; a value of 0 indicates that the simulations have not even begun to converge. Interestingly, the TIP3P distribution is by far the right-most, followed by OPC3, demonstrating that the 3-point models have achieved the most

convergence given a fixed amount of simulation time. It could thus be expected that 3-point water models reach convergence faster and require less simulation time than 4- and especially 5-point models. Moreover, with all three water models examined here, there is a discernible dependence between molecular weight and convergence–the higher the weight, the lower the A_c metric (S5 Fig), indicating that larger ligands require longer simulations to converge. Further still, there is significant agreement between the convergence estimates among the different water models (S5 Fig).

Another important property that can be expected to strongly affect the convergence in molecular simulations is the number of rotatable bonds (henceforth also referred to as rotors for brevity) a molecule has. In Fig 4, we show a breakdown of the A_c convergence metric and molecular weight by the number of rotatable bonds.

We see that as the number of rotors increases, the A_c distributions shift downward to lower values, whereas the weight distributions move to the right, indicating that molecules with more rotors tend to have larger molecular weights and tend to exhibit lower convergence scores. With all three water models examined here, nearly all molecules with 8 or more rotatable bonds have an A_c value below 0.4, whereas nearly all ligands with 0 or 1 rotor have an A_c value above 0.4.

We uncover and report a connection between molecular size and calculation error. We focus on molecular weight because it is a direct measure of how much there is to describe in a molecule. However, molecular weight is not the only important factor that should be examined, the number of rotors perhaps being the next logical choice. However, we note that the number of rotatable bonds, being an integer, is a far more coarse descriptor than molecular weight. Moreover, significant chemical complexity can be introduced with few or no rotatable bonds, e.g. with large aromatic or (spiro)cyclic systems. For example, Fig 4 shows that there are compounds with 0 or 1 rotatable bonds but with large molecular weight (around the 400 g/mol region, blue rectangles) and that most of these tend to have lower A_c values than compounds with the same number of rotors but lower weight (the regions above and to the left of the blue rectangles in Fig 4). Moreover, there are ligands with considerably more rotatable bonds and lower weight that have higher convergence metrics (these are the ligands in the 2 or 3, 4 or 5, and 6 or 7 rotors panels lying above the level of the blue rectangles). Finally, Fig 4 shows that there are ligands with 6 or 7 rotatable bonds that also have molecular weights in the 400 g/mol region (red rectangles in the figure). These have similar convergence values to the ligands with 0 or 1 rotor in the 400 g/mol region (blue rectangles) indicating that molecular weight is a better predictor of convergence than the number of rotatable bonds. Indeed, molecular weight is one of the few descriptors that begin to approach significance in correlations to the magnitude of the calculated error (R = 0.38, R = 0.36, and R = 0.43 for the TIP3P, OPC, and OPC3 models, respectively), i.e. the higher the weight, the larger the error in the calculated result, whereas the number of rotors has no correlation to error (R < 0.1 for all three water models). Interestingly, the descriptor that exhibited the highest correlation to the magnitude of the error is the number of heteroatoms–R = 0.53 with TIP3P, R = 0.49 with OPC3; the correlation is much lower for OPC (R = 0.30).

No single descriptor is likely to account for a majority in the variance in the results. Moreover, we find that more sophisticated surface area descriptors such as PEOE_VSA14 and SlogP_VSA3 [71] are among the most highly correlated to error in HFE calculations and that they correlate to error more strongly than simpler descriptors such as total polar area or total molecular surface. PEOE_VSA and SlogP_VSA are the van der Waals atomic surfaces in a molecule that fall within a specific range of partial charge [72] and logP (the octanol/water partition coefficient) values, respectively; they are specific subdivisions of the molecular surface. Our feature engineering efforts, where we normalize properties by molecular weight, also

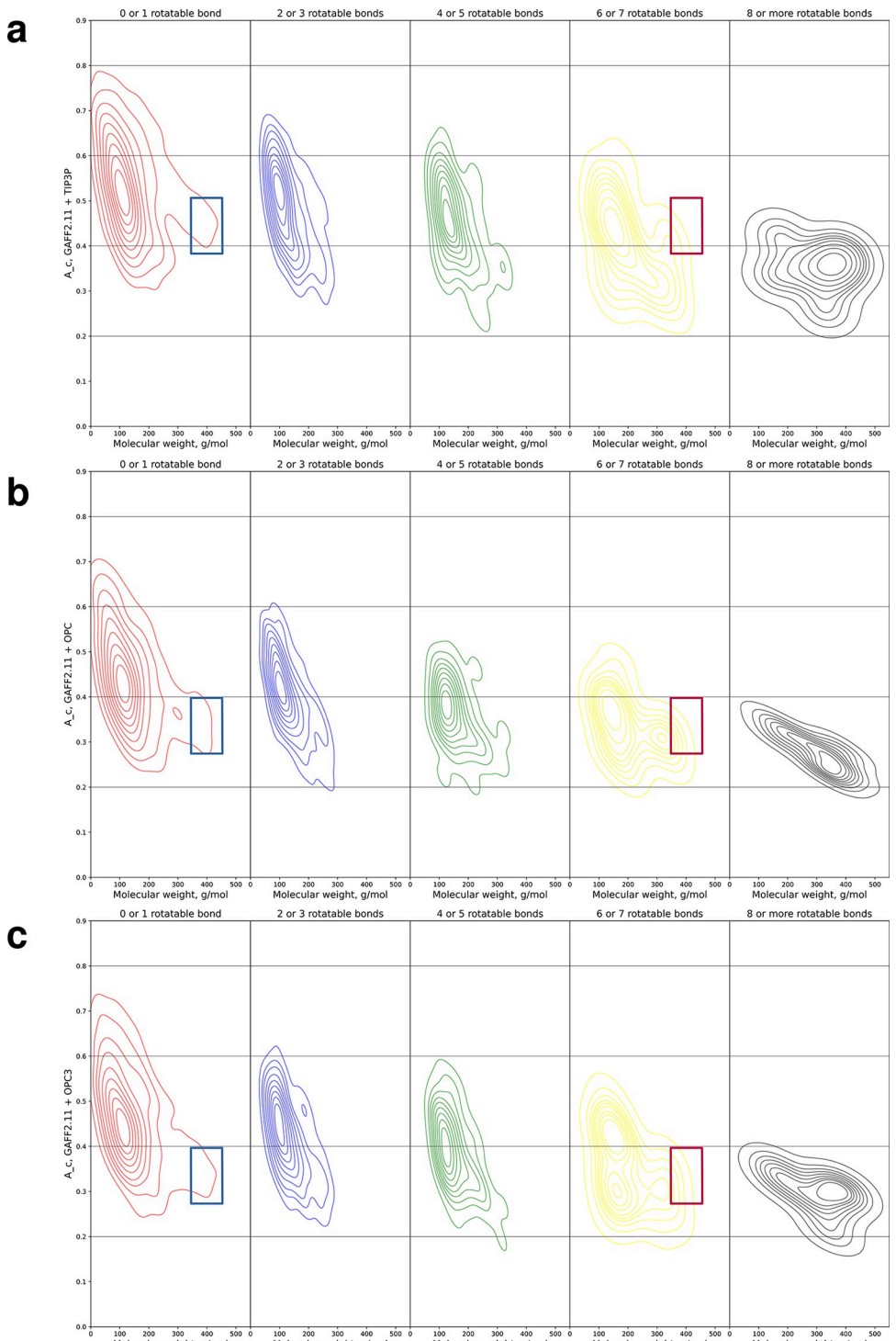

**Fig 4. A_c convergence metric and molecular weight distributions broken down by number of rotatable bonds. a** Bivariate kernel density estimate plot of the A_c convergence metric for GAFF2.11 with the TIP3P water model plotted against molecular weight. Ligands have been grouped in 5 separate panels based on the number of rotatable bonds– 0 or 1, 2 or 3, 4 or 5, 6 or 7, and 8 or more. For ligands with 0 or 1 or 6 or 7 rotatable bonds, the molecular weight regions around 400 g/mol have been highlighted with blue and red rectangles, respectively. **b** Bivariate kernel density estimate plot of the A_c convergence metric for GAFF2.11 with the OPC water model plotted against molecular weight. Ligands have been grouped in 5 separate panels based on the number of rotatable bonds– 0 or 1, 2 or 3, 4 or 5, 6 or 7, and 8 or more. For ligands with 0 or 1 or 6 or 7 rotatable bonds, the molecular weight regions around 400 g/mol have been

highlighted with blue and red rectangles, respectively. **c** Bivariate kernel density estimate plot of the A_c convergence metric for GAFF2.11 with the OPC3 water model plotted against molecular weight. Ligands have been grouped in 5 separate panels based on the number of rotatable bonds– 0 or 1, 2 or 3, 4 or 5, 6 or 7, and 8 or more. For ligands with 0 or 1 or 6 or 7 rotatable bonds, the molecular weight regions around 400 g/mol have been highlighted with blue and red rectangles, respectively.

proved productive in that some of the resulting features also showed significant (anti)correlation to HFE errors. One such example is VSA_Estate3 –the electrotopological state [70] within a specific range of van der Waals surface area values–normalized by molecular weight (VSA_Estate3/MolWt, all descriptors are available in S1 File). One could envisage further feature engineering schemes, e.g. normalizing by the number of rotatable bonds or even molecular weight divided by the number of rotatable bonds, to name but two. With the present report, we hope to stimulate further investigations into this matter and, more broadly, to stimulate deeper analysis into results coming out of computational chemistry pipelines. To this end, we make our data fully and freely available (S1 File).

Performing 3 more independent replicas with the TIP3P water model yielded nearly identical results–the 4 replicas agree with each other up to an $R^2$ of 0.99 and three of the four replicas have the same $R^2$ with the experimental HFEs (0.84). Only replica 3 has an $R^2$ of 0.83, indicating that few new ligand conformations are sampled by the additional replicas or that if new conformations are sampled, these are isoenergetic with previously sampled conformations. While it is generally recognized that multiple short simulations are more efficient in covering conformational space than one long simulation [73], our results demonstrate that this general rule has to be applied judiciously–there exists a threshold below which additional sampling becomes inefficient, failing to reach new energy basins from the conformational landscape, all the while adding computational costs.

We emphasize that our goal here is not to draw definitive conclusions about the different force fields and water models and recommend one combination over another but to demonstrate how they should be rigorously and thoroughly interrogated and validated. Ideally, new water models, force fields or versions of force fields should demonstrate appreciably improved performance in terms of error distributions, both absolute and relative, centered around x = 0 kcal/mol and x = 100% (perfect predictions), respectively, being taller and narrower than their predecessor or the previous gold standard, rather than merely improving parameters for one group or another or marginally improving $R^2$ in benchmarking. This does not appear to be the case for GAFF2.11 and 1.81, at least when using TIP3P water (S3 Fig). However, we reserve judgement until sufficient sampling can be attained, and merely suggest how this matter should be interrogated.

While our simulations are fairly short, our results are nearly identical to those of Matos et al. [74], who used a very similar protocol to our own with 20 λ windows and 5 ns of sampling per window– 20-fold greater than what we have used. Running nearly 20 times more computing with TIP3P and GAFF1.7 achieves a modest improvement in accuracy–$R^2$ = 0.87 (from 100 ns of total sampling per ligand) vs 0.84 from our results (5.25 ns of total sampling); RMSE = 1.53 ± 0.07 kcal/mol (from 100 ns of sampling) vs 1.78 ± 0.05 kcal/mol from our work (5.25 ns of sampling). Indeed, our results from 5.25 ns of sampling per ligand are virtually identical to their results from 100 ns of sampling ($R^2$ = 0.97). Further details become evident when comparing their results to the ones we report here with a plot of errors as a function of molecular weight (Fig 5).

We see that in the low-weight region of the plot, the results from their ~100 ns simulations lie closer to the x = 0 kcal/mol error axis (perfect prediction) than our results from ~5 ns of sampling. However, in the high-weight region, the additional sampling has not always

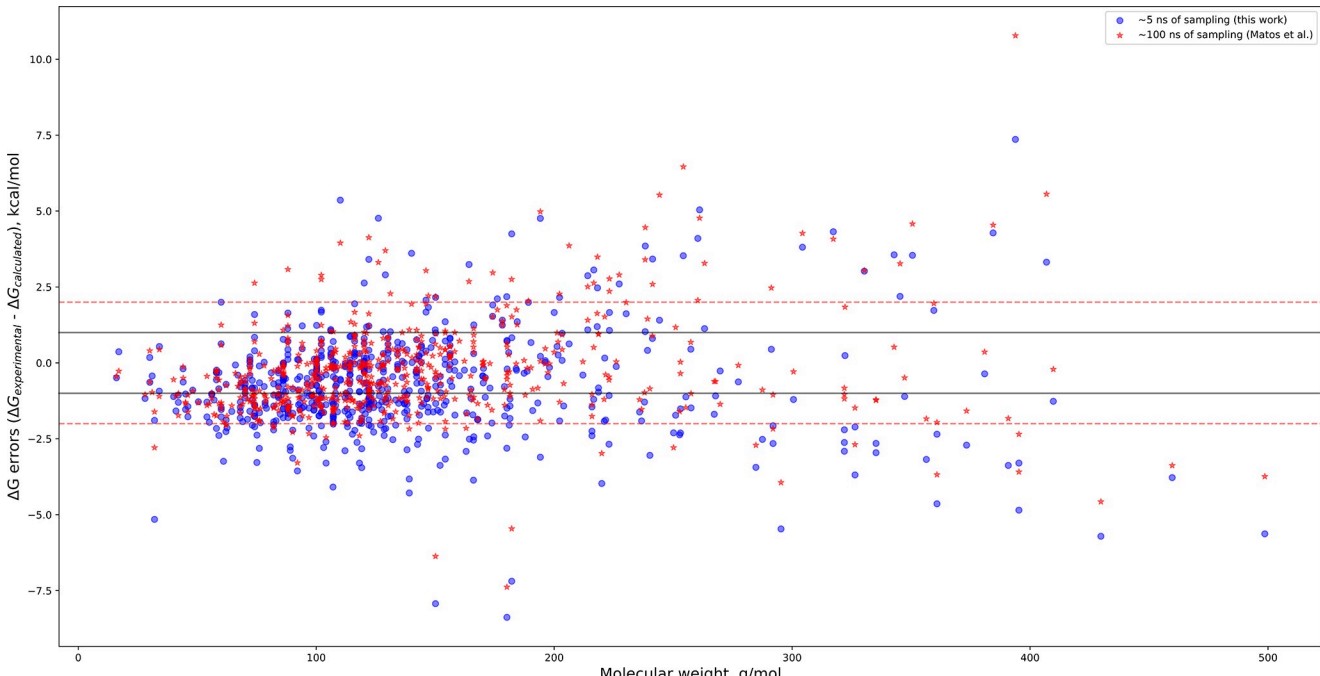

**Fig 5. Hydration free energy prediction errors versus molecular weight.** HFE errors ($\Delta G_{experimental} - \Delta G_{calculated}$) from this work and from Matos et al. [74] plotted against molecular weight. The regions between the horizontal solid and dashed lines indicate the location of predictions within 1 and 2 kcal/mol of experimental values, respectively.

produced more accurate results, as is evident from Fig 5. For some of the ligands, the results from Matos et al. are less accurate then our own, despite the 20-fold increase in sampling time. Regrettably, Matos et al. do not provide an estimate for the convergence in their simulations; it would be very interesting to see how convergence scales with simulation time.

Given the fact that 100 ns of sampling are often not enough to achieve accurate results for larger ligands, a conservative estimate would posit a requirement of one microsecond of sampling for ligands with drug- or drug-like molecular weights, which could be loosely defined as 400 g/mol or more. For single-digit numbers of drug-like ligands, this amount of sampling is still trivial in terms of cost. For large sets of ligands, however, it becomes advantageous to turn to ML models, which also appear to have more favorable error profiles–SVM models exhibit a much narrower error distribution across the entire molecular mass range than TI-based estimates. Crucially, the key advantage of ML HFE models is that they exhibit no real dependence between accuracy and ligand molecular weight–the $\Delta G$ error distribution for the ML predictors does not broaden moving to the right, unlike the distribution for TI-based estimates.

## Conclusions

While herein we focus primarily on molecular weight and to a lesser extent on rotatable bonds, we merely view this work as laying out a blueprint for interrogating results from HFE calculations and in computational chemistry more broadly. That is to say that not only should novel descriptors [75] be explored but also feature engineering with existing descriptors. We examine molecular weight and rotatable bonds only as the first properties that should be looked at while interrogating results from HFE, ABFE or RBFE calculations; we encourage the reader to think carefully about what other properties might be highly relevant to their particular data set(s) and offer potential feature engineering pathways for the reader to explore–

normalization by weight, rotors, or both; other avenues could certainly prove fruitful. Further, we hope to inspire more rigorous validation of HFE and MD results in order to avoid situations where overall statistics look favorable, but that is only because the data sets being used are dominated by molecules with properties that are easy to (retro)predict. Fig 5 offers a striking example of this–the overwhelming majority of data points lie in the low weight region below 200 g/mol and have their HFEs predicted to a high degree of accuracy. This makes overall RMSE and $R^2$ values appear favorable, but overshadows the fact that HFE calculations seem to struggle beyond 350 or 400 g/mol–the very region they are most needed. The lack of negative data in published computational chemistry work, particularly ABFE and RBFE studies, is an even more egregious flaw we plan to address in a future publication. If computer-aided drug design is to truly usher in the next generation of pharmaceuticals, it has to go beyond the bay and sail the sea.

## Supporting information

**S1 Fig. Experimental and computed hydration free energies calculated with GAFF2.11 and the TIP3P, OPC, and OPC3 water models.** The density distributions and scatter plots between the four properties along with the corresponding $R^2$ values and equations of best fit are given.
(TIFF)

**S2 Fig. Experimental and computed hydration free energies calculated with GAFF2.11 and the TIP3P water model in four independent replicas.** The density distributions and scatter plots between the five properties along with the corresponding $R^2$ values and equations of best fit are given.
(TIFF)

**S3 Fig. Hydration free energy prediction errors. a** HFE error ($\Delta G_{experimental} - \Delta G_{calculated}$) distributions for GAFF versions 2.11 and 1.81 calculated with the TIP3P water model. The x = 0 kcal/mol axis indicates the position of perfect predictions; the regions between the vertical solid and dashed lines indicate the location of predictions within 1 and 2 kcal/mol of experimental values, respectively. **b** HFE relative error ($\Delta G_{calculated}*100/\Delta G_{experimental}$) distributions for GAFF versions 2.11 and 1.81 calculated with the TIP3P water model. The x = 100% axis indicates the position of perfect predictions; the x = 0% axis separates ligands whose calculated HFE is of the correct sign (right of the axis) from ligands whose HFE sign has been mispredicted (left of the axis).
(TIFF)

**S4 Fig. Convergence metric (A_c) distributions from the TIP3P, OPC, and OPC3 simulations with GAFF2.11.**
(TIFF)

**S5 Fig. Molecular weights for the FreeSolv ligands and convergence metrics (A_c) from their corresponding simulations with the TIP3P, OPC, and OPC3 water models using GAFF2.11.** The density distributions and scatter plots between the four properties along with the corresponding $R^2$ values and equations of best fit are given.
(TIFF)

**S1 File. A spreadsheet of all molecules used in this study.** For ease of comparison, the *parm* column gives the molecular ID from the corresponding Matos et al. paper [74], followed by the SMILES, name, the experimental HFE, the experimental uncertainty, the HFE values calculated by Matos et al. and their uncertainties, followed by our results. The *ti_A_c_tip3p_gaff2.11*

column contains the A_c convergence metric values with GAFF2.11 and TIP3P, the *ti_tip3p_-gaff2.11* column gives the calculated TI values, and the *ti_uncertainty_tip3p_gaff2.11* column contains our error estimates from the calculations. What follows are analogous columns for the calculations with GAFF2.11 and OPC3 and GAFF2.11 with OPC; the GAFF1.81 with TIP3P calculations, followed by the *RDKit* canonical SMILES, and the *RDKit* molecular descriptors (their definitions can be found in the *RDKit* documentation and the references therein), followed by the same descriptors normalized by molecular weight. Finally, the *tip3-p_errors* column lists the errors from the HFE calculations using GAFF2.11 and TIP3P, analogously for *opc3_errors* and *opc_errors*; *mobley_errors* contains the errors from the Matos et al. calculations; the last six columns give the A_c and HFE values from the 3 additional replicas with GAFF2.11 and TIP3P from this work. The author declares that no competing interests exist.
(CSV)

## Author Contributions

**Conceptualization:** Stefan M. Ivanov.

**Data curation:** Stefan M. Ivanov.

**Formal analysis:** Stefan M. Ivanov.

**Visualization:** Stefan M. Ivanov.

**Writing – original draft:** Stefan M. Ivanov.

**Writing – review & editing:** Stefan M. Ivanov.

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
