## [Decision Letter · Decision Letter 0]

18 Jun 2024

PONE-D-24-21031Calculated hydration free energies become less accurate with increases in molecular weightPLOS ONE

Dear Dr. Ivanov,

Thank you for submitting your manuscript to PLOS ONE. After careful consideration, we feel that it has merit but does not fully meet PLOS ONE’s publication criteria as it currently stands. Therefore, we invite you to submit a revised version of the manuscript that addresses the points raised during the review process. Your manuscript has been reviewed by three independent reviewers and they raised valuable questions which require further clarifications and should be reflected in the revised manuscript. I will be delighted to consider the revised version of the manuscript. 

We look forward to receiving your revised manuscript.

Kind regards,

Soumendranath Bhakat

Academic Editor

PLOS ONE

Journal Requirements:

2. Please note that PLOS ONE has specific guidelines on code sharing for submissions in which author-generated code underpins the findings in the manuscript. In these cases, we expect all author-generated code to be made available without restrictions upon publication of the work. 

Please review our guidelines at https://journals.plos.org/plosone/s/materials-and-software-sharing#loc-sharing-code and ensure that your code is shared in a way that follows best practice and facilitates reproducibility and reuse.

"Acknowledgments. This study is financed by the European Union-NextGenerationEU through the National Recovery and Resilience Plan of the Republic of Bulgaria, project № BG-RRP-2.004-0004-C01. The in silico calculations were performed at the Centre of Excellence for Informatics and ICT, supported by the Science and Education for Smart Growth Operational Program and co-financed by the European Union through the European Structural and Investment Funds (Grant No. BG05M2OP001-1.001-0003)."

Please note that funding information should not appear in the Acknowledgments section or other areas of your manuscript. We will only publish funding information present in the Funding Statement section of the online submission form. Please remove any funding-related text from the manuscript. 

5. In this instance it seems there may be acceptable restrictions in place that prevent the public sharing of your minimal data. However, in line with our goal of ensuring long-term data availability to all interested researchers, PLOS’ Data Policy states that authors cannot be the sole named individuals responsible for ensuring data access (http://journals.plos.org/plosone/s/data-availability#loc-acceptable-data-sharing-methods).

**Additional Editor Comments:**

Please provide Github link to scripts and data used in this study

Reviewers' comments:

Reviewer's Responses to Questions

**Comments to the Author**

1. Is the manuscript technically sound, and do the data support the conclusions?

Reviewer #1: Yes

Reviewer #2: Partly

Reviewer #3: Partly

2. Has the statistical analysis been performed appropriately and rigorously? 

Reviewer #1: Yes

Reviewer #2: No

Reviewer #3: No

3. Have the authors made all data underlying the findings in their manuscript fully available?

Reviewer #1: Yes

Reviewer #2: No

Reviewer #3: No

4. Is the manuscript presented in an intelligible fashion and written in standard English?

Reviewer #1: Yes

Reviewer #2: No

Reviewer #3: Yes

5. Review Comments to the Author

Reviewer #1: This manuscript focuses on the impact of molecular properties & choice of water model on hydration free energy – a key component in determination ligand binding free energy. The findings and discussion presented in this study add value to the area of computational drug discovery efforts. I recommend this manuscript for publication after the following comments have been addressed by the author.

1. The author has measured molecular descriptors using RDKit. However, the discussion on the measured descriptors is missing in the discussion. I would recommend adding the list of the descriptors. Did the author find any correlation between one or more of those descriptors and the RMSE in HFE? Did any of those descriptors show correlation with the errors in the sign of the HFE?

2. The results presented in this manuscript show that the 3-point water models outperform the 4-point water model. Could this finding be extended to charged molecules? Should the choice of water model be determined by the physicochemical property of the molecules? A comment on this aspect could be useful if the author has any insights obtained from their analysis.

3. While the author has presented an interesting observation regarding the impact of molecular weight on the accuracy of hydration free energy prediction, it will be useful to add a section on why the errors get larger with increase in molecular weight. Specifically, does it have any correlation with the accessible surface area and total volume of the molecule? Do linear molecules show a different trend than cyclic or branched molecules?

Reviewer #2: See Attached Document for formatted review. Plaintext version pasted below:

Review for :

“Calculated hydration free energies become less accurate with increases in molecular weight”

Summary: This article seeks to address and benchmark long-standing problem of how accurately computation is able to predict the hydration free energy (HFE) of drug-like molecules. By using an established dataset of experimentally measured HFE values, the FreeSolv database, the author uses AMBER MD engine to carry out free energy perturbation methods to compute HFE estimations. The author then generates an machine-learned (ML) model trained on this data and assess the performance of said model relative to HFE predictions. The analysis of accuracy focuses on some select molecular properties, such as molecular weight, but does not do a complete benchmarking of other potentially relevant properties. As such, the article could benefit from a greater in-depth discussion on the results, why certain features were chosen, and more discussion on HFE’s relevance to drug discovery. As discussed below, other parts require further contextualization, and the discussion section requires a major rewrite as it is a single large paragraph.

Major (required) revisions:

1. The introduction would benefit from a greater direct contextualization of HFE calculations and their relevance in drug discovery. While Free Energy Calculations (FECs) are known to be incredibly relevant to drug discovery and inhibitor improvement, the introduction does not make it clear that HFEs also have their own utility. It would be beneficial if the introduction provided additional citations and additional direct contextualization about where HFEs are valuable in the drug discovery process and what point they’re relevant. Obviously, solubility is an important component of the ligand design process, but there is no discussion about additional methods for measuring solubility like logS or how they are utilized.

2. The discussion requires a major rewrite. It is currently one singular large paragraph that spans multiple pages, making it incredibly difficult to read and parse. It is also not clear how the discussion points are connected to the data or are further speculation without the data. It will be important to separate the discussion to clearly demarcate both of those types of paragraphs.

3. Additional discussion on why comparing error to molecular weight and no other properties would benefit the manuscript. In general, I think it is known that HFEs will scale with respect to molecular weight, and with increased ligand size there will be increased parameters to consider and increased time to convergence with simulation-based methods. Thus, it seems consistent with expectation that HFE variance will increase per ligand which would be remedied by increased replicates.

4. Consistent with the above discussion that convergence would require additional sampling with larger molecular weight due to a variety of chemical factors, it would be useful to compare how different numbers of replicates impact this error while still achieving convergence. It would be incredibly useful to identify if there is a scaling for molecular weight, number of atoms/bonds, and the amount of sampling needed.

5. Lastly, it would be useful to provide additional contextualization, testing, and out-of-data comparisons for the Machine Learning model that was constructed. The model tested using these train-test splits never tested against another orthogonal dataset that might be out of distribution. Testing only on data within the dataset allows the model to learn more similar chemistries between the across the train-test split without any guarantee that the model would indeed be able to extrapolate to new chemical topologies.

6. Additionally, a more thorough of the train-test characterization would be useful. It is currently not clear whether the train-test splits contain similar chemical identities in both the training and test sides of the split, which would make it difficult to test how well the dataset is able to extrapolate to new chemistries. Given the importance of the FreeSolv dataset to the ML model, but its small size, it would be useful to do more thorough chemically driven splitting.

7. Given that this was an effort done using established open-source libraries on openly available databases, it would good for there to be an associated github for sharing the results of this data and their weights.

8. Given the interesting nature of the data and the results, a conclusion section would benefit the manuscript and greatly improve readability.

Minor revisions:

1. Citations are needed at the following points:

a. Page 2, line 31, “an HIV integrase inhibitor”

b. Page 2, line 32-33, “modeling and simulation have been instrumental in bringing aobut new therapeutic agents”

c. Page 4, line 86 and 87, “one-step approach” and “two-step approach”

d. Page 5, line 97, “despite its many known shortcomings”

e. Page 5, line 96, “if not the most widely used”

2. The following phrases are not clear and could benefit from rewording:

a. Page 3, line 43-44

3. Start a new paragraph at Page 5 line 108 for clarity.

4. The end of the introduction is riddled with sharp transitions between sentences – some transition words and rephrasing can improve the flow for the reader here.

5. Given the historically outdated nature of the Berendsen Barostat (see papers such as: https://doi.org/10.1016/j.molliq.2022.120116 and https://doi.org/10.1016/j.bbamem.2016.02.004), it would be useful to provide some context for why the Berendsen was used in this simulation over other barostat methods. Alternatively a characterization across different barostats would be useful to see.

6. It would be useful to provide increased contextualization in the text for building the 2D QSAR models

7. Page 8, line 166-167: It would be useful to provide a description what these parameters of zero-variance were that were removed from consideration, and what the other 284 descriptors were in the SI.

8. Page 8, line 169: Please clarify why a 70:30 ratio was chosen for train-test splitting (or provide a citation)

9. Figure would benefit from larger fonts and heading text to improve readability

10. Page 11, line 232: Perhaps a more quantitative description of what it means that the three modes are lying close to each other?

11. Page 14, line 301: Provide a rationale/citation for why 5.25 ns of production dynamics was used, or a more robust sampling was done.

12. Page 15, line 314 appears to have a typo – it should be referring to S5 Fig. if I’m reading correctly?

Reviewer #3: This work compares the experimental Hydration Free Energy (HFE) in the Free Solve dataset to a) values calculated using alchemical free energy methods with thermodynamic integration (TI) estimator and b) to an ML model trained on what I assume is the experimental data. The main conclusion is that the error in the prediction from the alchemical estimation increases with increasing molecular weight.

Overall

There has been considerable effort in producing a useful set of calculations on an important area of computational chemistry, I thank the authors for their efforts. The main points of criticism are that:

1. The implications and reasons for the lack of convergence of the HFE estimates are not adequately explored. This affects the validity of the conclusions drawn.

2. The inclusion of the ML analysis isn’t fully justified.

3. The discussion areas should be a lot more focused on the topic of the paper.

4. There should be more references to relevant work.

There is much here that is interesting and a refocusing of the analysis would be welcome to explore the convergence properties of these calculations.

Introduction

- The introduction is well written and gives an overview of computer aided drug discovery.

- I believe it is too long and not focused enough on the specific area covered by the work and fails to make the case for why this work is necessary.

- The have been numerous works looking at computational predictions of HFE e.g. (non-exhaustive list), The SAMPL challenges or https://doi.org/10.1021/acs.jcim.0c00600, https://pubs.acs.org/doi/abs/10.1021/acs.jcim.0c00285 which should be mentioned.

- There have also been many papers on ML methods for QSAR, see https://paperswithcode.com/sota/molecular-property-prediction-on-freesolv for a ‘leaderboard’ of methods on the Free Solv database.

- Other forcefields e.g., CHARMM small molecule forcefield and the recent Open Force Field, Sage 2.0, and Machine Learned forcefields were also not mentioned.

Methods

- The methods were mostly clearly explained with the following exceptions: The use of the TI estimator was not fully justified given the success of other estimators and potential drawbacks of TI, e.g., MBAR. See https://pubs.acs.org/doi/abs/10.1021/acs.jcim.0c00285, https://doi.org/10.1063/1.5041835, and https://doi.org/10.1021/acs.jced.7b00104.

- The QSAR variables would be better listed in the SI, rather than given as a reference. While implied, the training data was not explicitly identified as the experimental values. Given there is value and precedence in fitting ML models to ABFE data this should be explicitly mentioned.

- The normalization by the molecular weight, while not wrong, was not justified as inclusion of the molecular weight itself should account for the effect of molecular weight on the predictions and regression coefficients.

- The fitting procedure was rigorous but the hyperparameter turning curve should be given in the SI.

- The metric denoted as ‘relative error’ is not consistent with general use of that term (which would include subtraction by 1) please either subtract 1 from the values or use a different term.

Results

The results are generally well reported and clear.

- The vertical lines in figure 1 where not explained in the caption.

- The discussion of the distribution of prediction errors would benefit from being quantitative (using terms like, bias, standard deviation, kurtosis, etc.) rather qualitative (e.g. ‘TIP3P has the tallest and narrowest error distribution’).

- In line 236 please convert these values into percentages.

- Please clarify what you mean by (line 242) ‘not only do the errors become larger with increasing MW but…’ as the error distribution looks to have an approximate mean of 0 for larger weight. Your note of increasing range and kurtosis looks accurate though.

- It is hard to draw meaningful conclusions from figure 2 due to its format (scatter plot with different colours). Plotting an estimate of the mean and range / standard deviation etc. of the errors vs MW would be more informative. One could use a Gaussian process, LOWESS smoother or even categorise the molecular weight into ranges and plot box plots. LOWESS smoothers are available in the Seaborn in the regplot function.

- Figure 3 is quite confusing. It would be ideal if you could keep the format of the comparison the same as Figure 2.

Discussion

- I do not believe that the conclusions you draw here are adequately supported by the data. This is because the convergence of the free energy estimates using the FE method drops significantly with increasing molecular weight.

- It’s not clear why MW has been singled out as the factor influencing accuracy given that number of rotatable bonds must also be very influential.

- I would like to see an analysis of convergence wrt to MW stratified by number of rotatable bonds.

- I would also like some investigation into the reasons for the lack of convergence. It’s not clear what you mean by ‘independent’ replicas in line 324. If they are not different configurations, perhaps perform replicas on some of the least converged molecules with different starting configurations.

- Comparisons between forcefields and water models is valid but only with converged estimates.

- Line 329: sampling multiple short trajectories are only useful if the starting configurations are drawn from the equilibrium configurational distribution (see comment earlier as well).

- In line 362 you say it is not the purpose of the paper to draw definitive conclusions yet the title of the paper is very definitive.

- The discussion is very wide ranging and could do with being shortened and restricted to the main points of the paper.

- The inclusion of the SVM models in the study was not justified.

6. PLOS authors have the option to publish the peer review history of their article (what does this mean?). If published, this will include your full peer review and any attached files.

Reviewer #1: No

Reviewer #2: No

Reviewer #3: No

---

## [Author Response · Author response to Decision Letter 0]

24 Jul 2024

I thank the reviewers for their positive and constructive comments. Below, I address the concerns they have raised.

Reviewer #1: This manuscript focuses on the impact of molecular properties & choice of water model on hydration free energy – a key component in determination ligand binding free energy. The findings and discussion presented in this study add value to the area of computational drug discovery efforts. I recommend this manuscript for publication after the following comments have been addressed by the author.

1. The author has measured molecular descriptors using RDKit. However, the discussion on the measured descriptors is missing in the discussion. I would recommend adding the list of the descriptors. Did the author find any correlation between one or more of those descriptors and the RMSE in HFE? Did any of those descriptors show correlation with the errors in the sign of the HFE?

I have made the complete set of descriptors and calculated HFEs fully and freely available with the supplementary information to this paper. As suggested by the reviewer, I have explored individual descriptors in the Results and Discussion sections.

2. The results presented in this manuscript show that the 3-point water models outperform the 4-point water model. Could this finding be extended to charged molecules? Should the choice of water model be determined by the physicochemical property of the molecules? A comment on this aspect could be useful if the author has any insights obtained from their analysis.

Currently, alchemical HFE calculations such as the ones we describe here are not capable of handling charged ligands because during decoupling the system’s net charge would change. This introduces large errors of a complex nature; the relevant theory is described in more detail in https://doi.org/10.1063/1.4826261. This is why FreeSolv is composed mostly of molecules that have a charge of 0 near neutral pH.

3. While the author has presented an interesting observation regarding the impact of molecular weight on the accuracy of hydration free energy prediction, it will be useful to add a section on why the errors get larger with increase in molecular weight. Specifically, does it have any correlation with the accessible surface area and total volume of the molecule? Do linear molecules show a different trend than cyclic or branched molecules?

I thank the reviewer for this comment, as I feel it is particularly productive. It also aligns with some of the comments made by the other reviewers. Molecular surface descriptors have been explored in more depth in the paper and they do indeed show correlation to the errors in the HFE calculations. Interestingly, the more sophisticated RDKit surface descriptors such as PEOE_VSA and SlogP_VSA have higher correlations than simple descriptors such as total polar surface or total surface area. 

Reviewer #2: See Attached Document for formatted review. Plaintext version pasted below:

Review for :

“Calculated hydration free energies become less accurate with increases in molecular weight”

Summary: This article seeks to address and benchmark long-standing problem of how accurately computation is able to predict the hydration free energy (HFE) of drug-like molecules. By using an established dataset of experimentally measured HFE values, the FreeSolv database, the author uses AMBER MD engine to carry out free energy perturbation methods to compute HFE estimations. The author then generates an machine-learned (ML) model trained on this data and assess the performance of said model relative to HFE predictions. The analysis of accuracy focuses on some select molecular properties, such as molecular weight, but does not do a complete benchmarking of other potentially relevant properties. As such, the article could benefit from a greater in-depth discussion on the results, why certain features were chosen, and more discussion on HFE’s relevance to drug discovery. As discussed below, other parts require further contextualization, and the discussion section requires a major rewrite as it is a single large paragraph.

Major (required) revisions:

1. The introduction would benefit from a greater direct contextualization of HFE calculations and their relevance in drug discovery. While Free Energy Calculations (FECs) are known to be incredibly relevant to drug discovery and inhibitor improvement, the introduction does not make it clear that HFEs also have their own utility. It would be beneficial if the introduction provided additional citations and additional direct contextualization about where HFEs are valuable in the drug discovery process and what point they’re relevant. Obviously, solubility is an important component of the ligand design process, but there is no discussion about additional methods for measuring solubility like logS or how they are utilized.

As suggested by the reviewer, I have added an entire paragraph to the Introduction contextualizing HFEs and their relevance in drug design.

2. The discussion requires a major rewrite. It is currently one singular large paragraph that spans multiple pages, making it incredibly difficult to read and parse. It is also not clear how the discussion points are connected to the data or are further speculation without the data. It will be important to separate the discussion to clearly demarcate both of those types of paragraphs.

As suggested by the reviewer, the Discussion has undergone a major rewrite making it shorter and more focused on the work presented here. This comment also nicely aligns with comments by the other reviewers; I have taken this opportunity to add discussion on some of the descriptors to the Discussion section.

3. Additional discussion on why comparing error to molecular weight and no other properties would benefit the manuscript. In general, I think it is known that HFEs will scale with respect to molecular weight, and with increased ligand size there will be increased parameters to consider and increased time to convergence with simulation-based methods. Thus, it seems consistent with expectation that HFE variance will increase per ligand which would be remedied by increased replicates.

I thank the reviewer for this comment as it is particularly constructive. Indeed, my intention for this manuscript was to notify the community of the issue I report and to inspire more investigations into the connection between HFE results (and computational chemistry results in general) and molecular properties. I am well aware that molecular weight is far from the whole story. It is thanks to this comment that I realized that that is never explicitly stated in the text; it is not even suggested. Therefore, I have explicitly stated in the manuscript that it only aims to inspire further analysis; the paper does not claim to be comprehensive. Indeed, this subject can likely fill many more papers-worth of material, likely many books-worth. Here, I have included analysis and discussion on some of the more salient features – weight, number of rotatable bonds, and a few surface area descriptors. 

4. Consistent with the above discussion that convergence would require additional sampling with larger molecular weight due to a variety of chemical factors, it would be useful to compare how different numbers of replicates impact this error while still achieving convergence. It would be incredibly useful to identify if there is a scaling for molecular weight, number of atoms/bonds, and the amount of sampling needed.

The reviewer is quite correct to point out that it would be very interesting to see how convergence scales with different factors. Sadly, Matos et al. have not included an estimate for the convergence metric from their 100 ns simulations in the supplementary information of their paper. Therefore, we cannot estimate how convergence scales with simulation time. As for scaling with molecular weight, supplementary figure 5 contains such scatter plots where the R2 between weight and the scaling parameter is around 0.30. However, at this point I am reluctant to make any major claims other than to say there certainly is a relationship. Quantifying it more precisely is the subject of future work, both by myself and other authors. That and the other questions the reviewer poses here would have to be the subject of future investigations as more compute time is presently not available.

5. Lastly, it would be useful to provide additional contextualization, testing, and out-of-data comparisons for the Machine Learning model that was constructed. The model tested using these train-test splits never tested against another orthogonal dataset that might be out of distribution. Testing only on data within the dataset allows the model to learn more similar chemistries between the across the train-test split without any guarantee that the model would indeed be able to extrapolate to new chemical topologies.

I address this comment together with the following one, see below.

6. Additionally, a more thorough of the train-test characterization would be useful. It is currently not clear whether the train-test splits contain similar chemical identities in both the training and test sides of the split, which would make it difficult to test how well the dataset is able to extrapolate to new chemistries. Given the importance of the FreeSolv dataset to the ML model, but its small size, it would be useful to do more thorough chemically driven splitting.

The machine learning section is used only as a story-telling device to introduce the chemical descriptors that will be used to analyze the outcomes from the HFE calculations. Therefore, ML is not the focus of the work. 

7. Given that this was an effort done using established open-source libraries on openly available databases, it would good for there to be an associated github for sharing the results of this data and their weights.

All descriptors and HFE results are now included in the supplementary information of this paper for the computational chemistry community to use and analyze. Again, I hope to inspire more papers of this sort, potentially deriving better descriptors, models, and hopefully force fields.

8. Given the interesting nature of the data and the results, a conclusion section would benefit the manuscript and greatly improve readability.

As suggested by the reviewer, a Conclusions section has been added where I explicitly state the goals of this work and its implications.

Minor revisions:

1. Citations are needed at the following points:

a. Page 2, line 31, “an HIV integrase inhibitor”

b. Page 2, line 32-33, “modeling and simulation have been instrumental in bringing aobut new therapeutic agents”

c. Page 4, line 86 and 87, “one-step approach” and “two-step approach”

d. Page 5, line 97, “despite its many known shortcomings”

e. Page 5, line 96, “if not the most widely used”

As requested by the reviewer, all of the above references have been added.

2. The following phrases are not clear and could benefit from rewording:

a. Page 3, line 43-44

This has been simplified.

3. Start a new paragraph at Page 5 line 108 for clarity.

As requested by the reviewer, this is now in a separate paragraph.

4. The end of the introduction is riddled with sharp transitions between sentences – some transition words and rephrasing can improve the flow for the reader here.

The introduction has been rephrased slightly to improve readability.

5. Given the historically outdated nature of the Berendsen Barostat (see papers such as: https://doi.org/10.1016/j.molliq.2022.120116 and https://doi.org/10.1016/j.bbamem.2016.02.004), it would be useful to provide some context for why the Berendsen was used in this simulation over other barostat methods. Alternatively a characterization across different barostats would be useful to see.

Here, I use the Berendsen barostat simply because it is a popular choice and has been a part of our pipeline for a long time. I cannot make any claims about what the results would look like with other barostats, other than to say that I do not expect them to change much. Force fields should be the major determinant of the quality of the calculated results, not the barostat. After all, force fields are the field that is undergoing extensive development, barostats and thermostats – not so much. Presumably, barostats and thermostats are mature enough.

6. It would be useful to provide increased contextualization in the text for building the 2D QSAR models

Again, I do not want to place too much emphasis on the ML section; my primary focus is on the analysis of the HFE results.

7. Page 8, line 166-167: It would be useful to provide a description what these parameters of zero-variance were that were removed from consideration, and what the other 284 descriptors were in the SI.

All descriptors are now available in the SI of the paper.

8. Page 8, line 169: Please clarify why a 70:30 ratio was chosen for train-test splitting (or provide a citation)

70:30 is just a personal preference. Most often researches use 75:25. Indeed, this is the default in many packages, including scikit-learn, and often people use it without even realizing it. I tend to dislike it somewhat because I feel it artificially inflates results – if there is too much data in the training set and too little data in the test set, results may come out looking good simply because there is too little test data to make an error on. However, there is no split that is set in stone, it is entirely up to the user to choose the ratio. 

9. Figure would benefit from larger fonts and heading text to improve readability

10. Page 11, line 232: Perhaps a more quantitative description of what it means that the three modes are lying close to each other?

While it would be straightforward to add exact numbers, I believe the text is already fairly heavy on numbers and data. Rather, the goal here is to guide the reader and help them interpret the data rather than to give exact details about the distributions.

11. Page 14, line 301: Provide a rationale/citation for why 5.25 ns of production dynamics was used, or a more robust sampling was done.

The rationale here is quite simple – this is the most I could afford. What is notable is that results differ marginally from 100 ns of sampling, as in the Matos paper (see Fig 5). I stress that the point of the paper is NOT that I can reproduce their results but that my results and their results clearly show a sharp deterioration of performance for larger, more complex ligands. This is something the community will need to acknowledge and address going forward. 

12. Page 15, line 314 appears to have a typo – it should be referring to S5 Fig. if I’m reading correctly?

Indeed, the reviewer is correct, this should be S5 Fig. I thank the reviewer for pointing this out.

Reviewer #3: This work compares the experimental Hydration Free Energy (HFE) in the Free Solve dataset to a) values calculated using alchemical free energy methods with thermodynamic integration (TI) estimator and b) to an ML model trained on what I assume is the experimental data. The main conclusion is that the error in the prediction from the alchemical estimation increases with increasing molecular weight.

Overall

There has been considerable effort in producing a useful set of calculations on an important area of computational chemistry, I thank the authors for their efforts. The main points of criticism are that:

1. The implications and reasons for the lack of convergence of the HFE estimates are not adequately explored. This affects the validity of the conclusions drawn.

As requested by reviewers 2 and 3, more discussion on convergence has been added.

2. The inclusion of the ML analysis isn’t fully justified.

This comment aligns with one of the comments from reviewer #2. As I have noted in my response to reviewer #2, the ML section is used only as a story-telling device to introduce the chemical descriptors that will be used to analyze the outcomes from the HFE calculations. Again, ML is not the focus of the work. 

3. The discussion areas should be a lot more focused on the topic of the paper.

This comment also nicely aligns with comments from the other reviewers

---

## [Decision Letter · Decision Letter 1]

23 Aug 2024

Calculated hydration free energies become less accurate with increases in molecular weight

PONE-D-24-21031R1

Dear Dr. Ivanov,

We’re pleased to inform you that your manuscript has been judged scientifically suitable for publication and will be formally accepted for publication once it meets all outstanding technical requirements.

Kind regards,

Soumendranath Bhakat

Academic Editor

PLOS ONE

Additional Editor Comments (optional):

Dear Dr. Ivanov,

After carefully reviewing the reviewers comments, I am glad to accept your paper PONE-D-24-21031R1 for publication in PLOS ONE. Thanks for your patience during this process.

Best regards,

Soumendranath Bhakat

Reviewers' comments:

Reviewer's Responses to Questions

**Comments to the Author**

1. If the authors have adequately addressed your comments raised in a previous round of review and you feel that this manuscript is now acceptable for publication, you may indicate that here to bypass the “Comments to the Author” section, enter your conflict of interest statement in the “Confidential to Editor” section, and submit your "Accept" recommendation.

Reviewer #1: All comments have been addressed

Reviewer #2: All comments have been addressed

2. Is the manuscript technically sound, and do the data support the conclusions?

Reviewer #1: Yes

Reviewer #2: Yes

3. Has the statistical analysis been performed appropriately and rigorously? 

Reviewer #1: Yes

Reviewer #2: Yes

4. Have the authors made all data underlying the findings in their manuscript fully available?

Reviewer #1: Yes

Reviewer #2: (No Response)

5. Is the manuscript presented in an intelligible fashion and written in standard English?

Reviewer #1: Yes

Reviewer #2: Yes

6. Review Comments to the Author

Reviewer #1: The author has satisfactorily addressed all the questions. Therefore, I recommend the manuscript for publication.

Reviewer #2: The author has addressed all comments in my previous review and revised the manuscript appropriately.

7. PLOS authors have the option to publish the peer review history of their article (what does this mean?). If published, this will include your full peer review and any attached files.

Reviewer #1: No

Reviewer #2: No

---

## [Editor Report · Acceptance letter]

10 Sep 2024

PONE-D-24-21031R1 

PLOS ONE

Dear Dr. Ivanov, 

I'm pleased to inform you that your manuscript has been deemed suitable for publication in PLOS ONE. Congratulations! Your manuscript is now being handed over to our production team.

Kind regards, 

on behalf of

Dr. Soumendranath Bhakat 

Academic Editor

PLOS ONE